# Transient Receptor Potential Ankyrin 1 (TRPA1) Channel as a Sensor of Oxidative Stress in Cancer Cells

**DOI:** 10.3390/cells12091261

**Published:** 2023-04-26

**Authors:** Francesco Moccia, Daniela Montagna

**Affiliations:** 1Laboratory of General Physiology, Department of Biology and Biotechnology “L. Spallanzani”, University of Pavia, 27100 Pavia, Italy; 2Department of Sciences Clinic-Surgical, Diagnostic and Pediatric, University of Pavia, 27100 Pavia, Italy; d.montagna@unipv.it; 3Pediatric Clinic, Foundation IRCCS Policlinico San Matteo, 27100 Pavia, Italy

**Keywords:** cancer, reactive oxygen species, hydrogen peroxide, Transient Receptor Potential Ankyrin 1, Ca^2+^ signaling, nuclear factor erythroid 2-related factor 2, antioxidant defense, apoptosis

## Abstract

Moderate levels of reactive oxygen species (ROS), such as hydrogen peroxide (H_2_O_2_), fuel tumor metastasis and invasion in a variety of cancer types. Conversely, excessive ROS levels can impair tumor growth and metastasis by triggering cancer cell death. In order to cope with the oxidative stress imposed by the tumor microenvironment, malignant cells exploit a sophisticated network of antioxidant defense mechanisms. Targeting the antioxidant capacity of cancer cells or enhancing their sensitivity to ROS-dependent cell death represent a promising strategy for alternative anticancer treatments. Transient Receptor Potential Ankyrin 1 (TRPA1) is a redox-sensitive non-selective cation channel that mediates extracellular Ca^2+^ entry upon an increase in intracellular ROS levels. The ensuing increase in intracellular Ca^2+^ concentration can in turn engage a non-canonical antioxidant defense program or induce mitochondrial Ca^2+^ dysfunction and apoptotic cell death depending on the cancer type. Herein, we sought to describe the opposing effects of ROS-dependent TRPA1 activation on cancer cell fate and propose the pharmacological manipulation of TRPA1 as an alternative therapeutic strategy to enhance cancer cell sensitivity to oxidative stress.

## 1. Introduction

Reactive oxygen species (ROS) comprise a group of highly reactive oxygen-containing molecules, including the non-radical, hydrogen peroxide (H_2_O_2_), the free-radicals, hydroxyl radical (OH^•^), superoxide anion (O_2_^•^), peroxides (RO^•^), and oxides of nitrogen (RO^•^) [1,2,3,4]. Aberrant redox homeostasis represents a hallmark of cancer cells, since moderate ROS (e.g., 10–50 µM H_2_O_2_ and O_2_^•^ in the nanomolar range) stimulate cell transformation, hyperproliferation, invasion, metastasis, and angiogenesis (Figure 1) [1,2,3,5,6,7,8]. The rate of basal ROS production in cancer cells is enhanced by multiple mechanisms [2,9], such as metabolic disturbances, adaptation to hypoxia, oncogene activation, and loss of tumor suppressors. However, excessive ROS levels (e.g., >100 µM H_2_O_2_; O_2_^•^ in the micromolar range; ≈30 µM OH^•^) could impair tumor development and spread by triggering cell apoptosis, ferroptosis, or senescence [8,10,11,12,13]. Therefore, cancer cells cope with oxidative stress by exploiting a sophisticated network of antioxidant defense mechanisms [10,14]. Non-enzymatic small molecules directly scavenge ROS and comprise the endogenously synthesized glutathione (GSH), melatonin, and melanin, as well as the exogenously derived vitamin C, vitamin E, and β-carotene [10]. Noteworthily, GSH expression is up-regulated in a variety of cancer cell types [15], thereby increasing their antioxidant capacity. Enzymatic mechanisms include catalase (CAT), which degrades H_2_O_2_ to H_2_O and oxygen (O_2_), peroxiredoxins (PRXs), and glutathione peroxidases (GPXs), which reduce H_2_O_2_ to O_2_, and superoxide dismutases (SODs), which catalyze the conversion of O_2_^•^ to H_2_O and O_2_ (Figure 1) [10]. Six PRX isoforms have been described not only in the cytosol but also in multiple organelles, including mitochondria, the endoplasmic reticulum (ER), and peroxisomes, whereas eight GPX isoforms scavenge H_2_O_2_ in the cytosol and mitochondria [10,16]. Reduced thioredoxin (TRX) and reduced glutathione (GSH), respectively, serve as cofactors for PRXs- and GPXs-mediated reduction of H_2_O_2_ to H_2_O. Additionally, GSH is used by glutathione-S-transferases (GSTs) to detoxify reactive compounds generated by oxidative stress [10,16]. Also SODs are spatially distributed in different subcellular compartments to favor O_2_^•^ elimination: Cytoplasmic SOD (SOD-1 or Cu/Zn-SOD), mitochondrial SOD (SOD-2 or Mn-SOD), and extracellular SOD (SOD-3 or EC SOD) [17]. SOD-1 and SOD-2 rapidly dismutate O_2_ into H_2_O_2_, which is less reactive and is reduced to O_2_ and H_2_O_2_ by catalase or converted to H_2_O_2_ and oxidized glutathione by GPx [17].

The transcription factor, nuclear factor-erythroid 2 p45-related factor 2 (NRF2), is the master regulator of redox homeostasis in cancer cells [1,10,14]. NRF2 activity is finely regulated by kelch-like ECH-associated protein 1 (KEAP1) and the Cul3-based E3 ubiquitin ligase, which target NRF2 for proteosomal degradation. High ROS levels cause the oxidation of redox-sensitive cysteine residues in KEAP1, thereby preventing the physical interaction with and subsequent degradation of NRF2. The latter can in turn translocate into the nucleus and drive the expression of numerous antioxidant genes [10,16]. These include ROS-detoxifying enzymes, e.g., GPXs, GSTs, and PRXs, as well as the two subunits comprising the glutamate–cysteine ligase (GCL), i.e., the catalytic subunit (GCLC) and the modifier subunit (GCLM), which catalyzes the rate-limiting step in GSH biosynthesis [1,10,14]. Therefore, an alternative anticancer strategy could be designed by either increasing ROS production or reducing the antioxidant capacity of cancer cells [1,10,14,18].

Transient Receptor Potential Ankyrin 1 (TRPA1), the unique member of the mammalian TRPA sub-family, is a Ca^2+^-permeable, non-selective cation channel that is able to integrate thermal, mechanical, and chemical signals [19,20]. Among its multiple endogenous agonists, ROS are crucial to activate TRPA1 in many disorders featured by oxidative stress, including neuropathic pain, inflammation, osteoarthritis, migraine, postischemic dysesthesia, diabetes, and respiratory diseases [19,21,22]. TRPA1-dependent extracellular Ca^2+^ entry can support several cancer hallmarks, including hyperproliferation, survival against pro-apoptotic stimuli, and invasive behavior [16,23,24,25]. In addition, TRPA1-mediated depolarization of peripheral nociceptors is involved in cancer-induced bone pain and cancer-related neuropathic pain [26,27]. A recent series of studies showed that TRPA1 may also serve as a crucial sensor of redox signaling in cancer cells and that extracellular Ca^2+^ entry through TRPA1 can promote either cell survival [28,29,30] or cell death [31,32,33] in response to oxidative stress. Herein, we first summarize the current knowledge about the structure and gating mechanisms of TRPA1. Then, we briefly survey the contribution of TRPA1-mediated intracellular Ca^2+^ signals to cancer cell proliferation, migration, and angiogenesis. Finally, we discuss how the redox-sensing capability of TRPA1 could be used by certain cancer cells to engage a non-canonical antioxidant defense program, while ROS-dependent TRPA1 activation leads to intracellular Ca^2+^ overload, mitochondrial dysfunction, and apoptosis in other cancer types. Therefore, TRPA1 channels could represent a novel molecular target for anticancer therapy that could be exploited either to dampen the antioxidant capacity of malignant cells or to exacerbate their sensitivity to ROS signaling.

## 2. TRPA1: Molecular Structure, Biophysical Properties, and Pharmacological Sensitivity

The mammalian TRP superfamily of non-selective cation channels encompasses 28 members that are subdivided into 6 sub-families based on their sequence homology: Canonical (TRPC1-7), melastatin (TRPM1-8), vanilloid (TRPV1-6), ankyrin (TRPA1), polycystin (TRPP), and mucolipin (TRPML1-3). The TRPP sub-family consists of eight members, but only TRPP2, TRPP3, and TRPP5 function as ion channels [34,35]. TRPA1 is the sole member of the TRPA subfamily and has originally been detected in a subpopulation of Aδ- and C-fiber nociceptive sensory neurons, in which it can serve as a chemical, mechanical, and thermal nocisensor [19,20]. Subsequently, TRPA1 has been found in other cell types such as epithelial cells, fibroblasts, enterochromaffin cells, mast cells, melanocytes, odontoblasts, and β-cells of the Langerhans islets, which may serve as sensory cells and interact with adjoining nociceptors [19,20]. More recently, TRPA1 expression has been confirmed in the central nervous system [36] and in cancer cells [16]. Interestingly, in the dorsal root ganglion, TRPA1 channels are also located in endolysosomes, thereby contributing to mediating intracellular Ca^2+^ release [37].

### 2.1. The Molecular Structure of TRPA1

The *TRPA1* gene is located in band q21.13 of chromosome 8 in humans and consists of 73.635 bases and 29 exons [38]. The protein channel encoded by the *TRPA1* gene presents an estimated molecular weight of ~127 kDa and long cytosolic NH_2_- and COOH-terminal tails, which collectively account for ~80% of the total protein mass [38,39]. The functional TRPA1 channel protein results from the assembly of four subunits into a homotetramer through ‘domain-swap’ interactions [39]. The molecular architecture of the TRPA1 protein has been recently solved at a near-atomic resolution (~4 Å) by using single-particle electron cryo-microscopy [39]. As predicted by the cDNA sequence [38], each subunit consists of six transmembrane (S1–S6) domains and presents an extracellular re-entrant pore loop between S5 and S6 (Figure 2) [39]. The long NH_2_-terminal of TRPA1 protein houses the most extensive ankyrin repeat domain (ARD) of the TRP superfamily, which comprises 14–16 ankyrin repeats (Figure 2), each consisting of a ~33 amino acids-long α-helix-β-turn-α-helix motif [39]. The ARD is connected to TM1 via the pre-S1 region, containing some cysteine residues (e.g., Cys621, Cys641, and Cys665) that are critical for TRPA1 activation by electrophilic agonists (Figure 2) [39,40]. In addition, the proximal portion of the COOH-terminus contains two residues, i.e., Arg975 and Lys989 (Figure 2), which control the voltage-dependent activation of TRPA1 at highly depolarizing potentials (>+100 mV) [41]. The ion conduction pathway of the TRPA1 channel is featured by two major constrictions, or gates, that resemble those also identified in the central cavity of TRP Vanilloid 1 (TRPV1). The outer gate is contributed to by diagonally opposed Asp915 residues, which are 7 Å apart and control Ca^2+^ permeability (Figure 2). The inner gate is formed by two hydrophobic seals established by Ile957 and Val961, which narrow the funnel to ~4 Å and thereby constrain the permeation of rehydrated cations (Figure 2). The outer pore domain of TRPA1 contains two α-helices, with a string of acidic amino acids (Glu920, Glu924, and Glu930) in the second α-helix that is likely to serve as a negatively charged conduit to repel anions and attract cations (Figure 2) [39].

### 2.2. Biophysical Properties and Gating Mechanisms of TRPA1

TRPA1 is a non-selective cation channel that is permeable to both monovalent (e.g., Na^+^ and K^+^) and divalent (e.g., Mg^2+^) cations and may be open in the absence of exogenous stimulation [19,42,43]. In the presence of extracellular Ca^2+^, TRPA1 presents a single-channel conductance of ~65 pS and ~110 pS for negative and positive membrane potentials, respectively [44]. Non-stimulated TRPA1 channels display a pore diameter of ~11 Å, a monovalent cation permeability sequence of Rb^+^ > K^+^ > Cs^+^ > Na^+^ > Li^+^, high permeability for Ca^2+^ over Na (P_Ca_/P_Na_ ~6), and a fractional Ca^2+^ current of ~17% [42,45]. Stimulation with electrophilic agonists, such as mustard oil, enhances the P_Ca_/P_Na_ to ~9 and the fractional Ca^2+^ current to ~23% and changes the monovalent cation permeability sequence to Ca^2+^ > Ba^2+^ > Mg^2+^ > NH4^+^ > Li^+^ > Na^+^ > K^+^ > Rb^+^ > Cs^+^ [42,45]. In addition, agonist exposure can change the dimensions of the selectivity filter, thereby resulting in the progressive, but reversible, dilation of the channel pore (by 1–3 Å), which can become permeable to large organic cations, such as N-methyl-D-glucamine and the cationic dye Yo-Pro [19,45,46,47]. An alternative model proposed that the increase in permeability to large molecules does not reflect a change in channel permeability, but rather an elevation in the intracellular ion concentration [48].

### 2.3. Ca^2+^-Dependent Regulation of TRPA1 Activity

Interestingly, TRPA1 activity can be modulated by both extracellular and intracellular Ca^2+^ [36,44]. An increase in [Ca^2+^]_i_ could initiate and augment TRPA1-mediated inward currents elicited by several compounds, such as Δ^9^-tetrahydrocannabinol (THC) and allyl isothiocyanate (AITC) [49]. Similarly, elevating the extracellular Ca^2+^ concentration also potentiated TRPA1 channel activation and inactivation [50]. These observations led to a model according to which extracellular Ca^2+^ permeates the channel pore and thereafter regulates TRPA1 activity by binding to a site that is located within or very close to the pore [19]. The NH_2_ tail contains a putative EF-hand Ca^2+^-sensing domain that is located on ARD12 and is likely to mediate the Ca^2+^-dependent activation of TRPA1 via the residues Asp466, Leu474 and Asp477 (Figure 2) [19,36,42,51]. An additional Ca^2+^-binding site, which is involved in the Ca^2+^-dependent activation and inactivation of TRPA1, is located at the distal COOH-terminal region and formed by Glu1077 and Asp1080-Asp1082 (Figure 2) [52]. Intriguingly, calmodulin (CaM) may be associated with TRPA1 in the presence of Ca^2+^ via the physical interaction with a CaM-binding domain in the COOH-tail of TRPA1 (Figure 2) [53]. It has been proposed that CaM promotes TRPA1 inactivation following extracellular Ca^2+^ influx [53].

### 2.4. TRPA1 Activity Is Stimulated by Physical Stimuli: Voltage, Temperature and Membrane Deformation

The gating of TRPA1 channels can be modulated by multiple physical stimuli, including changes in membrane potential, temperature, and plasma membrane tension [19]. TRPA1 can be activated by highly depolarizing voltages (>+100 mV) and presents a half-maximal voltage (V_1/2_) of channel activation ranging between +90 mV and +170 mV [42,44]. As anticipated in Section 2.1, the voltage-dependent gating of TRPA1 depends on four positively charged residues, i.e., Lys969, Arg975, Lys988, and Lys989, which are located within the most proximal helix (H1) of the COOH-terminus (Figure 2) [41]. The voltage sensitivity of the channel is also sensitive to single alanine mutations of multiple residues located in the predicted helix that is centered around Lys1048 and Lys1052 [41]. Membrane depolarization could promote the physical interaction between the distal region and the more proximal voltage sensor, thereby resulting in a conformation change that causes TRPA1 activation [42]. However, the voltage-dependent gating of TRPA1 can be shifted towards more physiological membrane potentials by Ca^2+^ (see Section 2.3), electrophilic and non-electrophilic agonists, and cold or hot temperatures [19,36,42]. In accord, human TRPA1 is intrinsically sensitive to both noxious cold (<12 °C) and noxious (>43 °C) heat [54,55], although its thermosensitive profile may depend on the species [56]. Of note, the heat and cold responsiveness of the human TRPA1 (hTRPA1) is finely tuned by its redox environment [56]. ROS produced in response to dramatic changes in the local temperature could represent the driving force underlying the activation of hTRPA1 by noxious cold or hot temperatures [54]. A recent investigation proposed that the COOH-terminal region of hTRPA1 harbors temperature-sensitive modules that are allosterically coupled to the S5–S6 pore region and the S1–S4 transmembrane domain [55]. The temperature sensitivity of hTRPA1 can be modulated by the NH_2_-terminal ARD [57] and is lost under reducing conditions [54]. Finally, TRPA1 channels present mechanosensitive activity that has largely been observed in C fibers exposed to large membrane deformation and in other cell types involved in mechanosensation, such as sensory neurons and Merkel cells [19,58]. Emerging evidence indicates that hTRPA1 is intrinsically mechanosensitive: when reconstituted in artificial lipid bilayers, its single-channel current activity increased in response to an increase in the lipid tension stress [59]. Interestingly, the mechanosensitivity of the human TRPA1 also does not require the NH_2_-terminal ARD but is finely tuned by its redox state and is favored by a pro-oxidant environment [59].

### 2.5. TRPA1 As a Sensor of Redox Signaling

ROS and ROS metabolites, as well as reactive nitrogen species (RNS), are the main endogenous regulators of TRPA1 activity under physiological conditions [17,19,36,42,44]. Therefore, TRPA1 belongs to the class of TRP channels that serve as sensors of the cellular redox state [60] and include TRP Melastatin 2 (TRPM2) [61,62] and TRP Vanilloid 1 (TRPV1) [63,64]. TRPA1 can be directly activated by H_2_O_2_ [65,66], OH [66], the cyclopentenone prostaglandin 15-deoxy-delta(12,14)-prostaglandin J(2) [15d-PGJ(2)] [65], nitric oxide (NO) [67], and peroxynitrite (ONOO^−^), [68]. In addition, TRPA1 channels can be stimulated by multiple endogenous aldehydes that are produced in response to lipid peroxidation [4,69], such as 4-hydroxynonenal (4-HNE), 4-oxo-nonenal, and 4-hydroxyhexenal [43,65]. Similarly, TRPA1 activity is enhanced by nitro-oleic acid, which is produced during the nitration of plasma membrane phospholipids [70]. ROS increase the open probability of TRPA1 via covalent modifications of three cysteine residues, i.e., Cys621, Cys641, and Cys665, which are situated within the pre-TM1 region at the NH_2_-terminal of the channel protein (Figure 2) [71]. The exceptional reactivity of Cys621 towards electrophiles is facilitated by Lys620, whereas full TRPA1 activation by oxidative stress requires the covalent modification of Cys665 [71]. The combination of cryo-electron microscopy, which solved the full-length structure of hTRPA1 at ~a 4 Å resolution [39], and molecular modeling [72] suggested that the electrophilic reactive Cys621, Cys641, and C665 could form a ligand-binding pocket by coming in close proximity with each other [42]. The pre-TM1 region contains three additional cysteine residues, i.e., Cys619, Cys639, and Cys663, and to a lesser extent Lys708, that are not directly modified by ROS, but react to other electrophilic TRPA1 agonists, such as AITC [73]. The electrophilic activation of TRPA1 channels can also be facilitated by the disruption or formation of disulfide bonds between these, as well as other, cysteine residues at the NH_2_-terminal [42,72]. Finally, the TM core of TRPA1 protein presents cysteine and lysine residues, i.e., Cys727, Lys771, and Cys834, that are likely to be exposed to the lipid environment and could, therefore, be reactive to lipophilic electrophiles [36,39].

Quantitative analysis demonstrated that TRPA1 is the most highly redox-sensitive TRP channel and can, therefore, uniquely serve as a sensor of molecular oxygen (O_2_) [74]. Under normoxic conditions (~20% O_2_), TRPA1 activity is tonically inhibited through hydroxylation of Pro394 in the NH_2_-terminal ARD by prolyl hydroxylases (PHDs) [74], which function as the main O_2_ sensor of the cell and regulate the stability of hypoxia-inducible transcription factors (HIFs) [75]. The hydroxylation activity of PHDs is decreased upon a reduction in O_2_ concentration, thereby relieving the channel from inhibition and leading to TRPA1 activation under hypoxia [74]. In addition, hypoxia can induce the insertion of non-hydroxylated TRPA1 channels into the plasma membrane [74]. On the other hand, hyperoxia can activate TRPA1 through the O_2_-dependent oxidation of two cysteine residues at the NH_2_-terminal of the channel protein, i.e., Cys633 and Cys656 [74]. Additionally, although seemingly paradoxical, an increase in oxidative stress could occur even during hypoxia. In accord, electron transfer from ubisemiquinone to O_2_ at the Q0 site of the mitochondrial complex III dramatically enhances ROS generation in hypoxic cells [76]. Therefore, TRPA1 may also serve as a molecular sensor of the local changes in the O_2_ concentration [77].

## 3. TRPA1-Mediated Ca^2+^ Signals Support Tumorigenesis

Because of the versatility of its gating mechanisms, TRPA1 is uniquely suited to detect the increase in ROS production that supports neoplastic transformation and dissemination [16]. An additional feature of solid malignancies is represented by hypoxia, which is due to their abnormal vascular network, comprising leaky, highly disorganized, and compressed capillary vessels [78,79]. Therefore, the cancer microenvironment has been recognized as the ideal milieu to activate TRPA1 [16,26,60]. In the present section, we will briefly survey the mechanisms whereby extracellular Ca^2+^ entry via TRPA1 channels promotes cancer cell proliferation, survival, migration, and angiogenesis. In Section 4, we will specifically address how redox-sensitive TRPA1 activation could either engage non-canonical antioxidant defense programs or induce apoptosis in cancer cells. In Section 5, we focus on the pharmacology of TRPA1 channels and describe how stimulating or inhibiting TRPA1 activity could represent a novel therapeutic strategy to induce ROS-dependent cancer cell death.

Remodeling of the Ca^2+^ handling machinery, including multiple members of the TRP superfamily, contributes to many cancer hallmarks, such as aberrant proliferation, tissue invasion and metastasis, resistance to pro-apoptotic chemotherapeutics, and sustained angiogenesis [80,81,82,83,84,85,86,87,88]. Early work showed that the TRPA1 protein was upregulated in several cell lines and in tumor samples of human small-cell lung cancer (SCLC) [33]. TRPA1-mediated extracellular Ca^2+^ entry prevented starvation-induced SCLC cell apoptosis by recruiting extracellular signal-regulated kinases 1/2 (ERK 1/2) in an Src-dependent manner [33]. A parallel investigation demonstrated that TRPA1 expression on the plasma membrane of human lung adenocarcinoma A549 cells can be increased by inflammatory cytokines, such as interleukin (IL)-1α, IL-1β, and tumor necrosis factor α (TNFα) [89]. Similarly, TRPA1 was expressed in prostate cancer stromal cells expanded from different patients, but not in healthy primary cultured prostate epithelial cells [90]. This study showed that the antibacterial agent, triclostan, stimulated TRPA1 to mediate extracellular Ca^2+^ entry, thereby resulting in vascular endothelial growth factor (VEGF) secretion and prostate cancer cell proliferation [90]. In addition, VEGF could target adjacent endothelial cells to induce sprouting angiogenesis and favor prostate cancer vascularization [91]. Interestingly, TRPA1 protein was also largely expressed in prostate tumor-derived endothelial cells (PTECs), while it was absent in its normal counterpart [92]. TRPA1-mediated intracellular Ca^2+^ signals stimulated PTECs to migrate and assemble in capillary-like networks both in vitro and in vivo [92]. Additionally, the TRPA1 protein was expressed and mediated an increase in the intracellular Ca^2+^ concentration ([Ca^2+^]_i_) in human prostate cancer-associated fibroblasts (CAFs) [93]. In prostate CAFs, TRPA1 could be activated by the natural polyphenolic antioxidant, resveratrol, and induced the secretion of VEGF and hepatocyte growth factor (HGF), which in turn reduced resveratrol-induced apoptosis in co-cultured human prostate cancer cells [93]. TRPA1 was also detected in human pancreatic ductal adenocarcinoma cells (PDACs), which displayed higher levels of TRPA1 mRNA expression as compared to non-neoplastic cells [24]. TRPA1 activity, both under basal conditions and in the presence of the electrophilic agonist AITC, reduced PDAC cell migration and caused changes in cell cycle progression, i.e., induced a shift from G0/G1 to a sub-G1 phase [24]. TRPA1 could also regulate PDAC cell motility in a flux-independent manner, i.e., without the requirements for extracellular Ca^2+^ entry, as recently suggested for other TRP channels [94,95] and ligand-gated ion channels [96,97,98]. In agreement with this hypothesis, TRPA1 can physically associate with the fibroblast growth factor receptor 2 (FGFR2) via its NH_2_-terminal ARD and thereby stimulate lung adenocarcinoma (LUAD) progression and metastatic spreading in a Ca^2+^-independent manner [99]. A subsequent report, however, showed that FGFR2 expression is rather low in TRPA1-expressin lung cancer cells and that the TRPA1–FGFR2 interaction is likely to be a rare event in LUAD [29]. Therefore, TRPA1 can contribute to tumorigenesis, although its effect can vary depending on the cancer type, e.g., TRPA1-mediated Ca^2+^ signals stimulate and inhibit migration in PTECs and PDACs, respectively, as discussed above. TRPA1 expression and functional activity have also been reported in human uveal melanoma 92.1 cells [25], human neuroblastoma IMR-32 cells [100], and human oral squamous cell carcinoma (OSCC) samples [23]. Certainly, the validation of TRPA1 as a novel molecular target for anticancer strategies would benefit from a wider knowledge of the impact of TRPA1-mediated Ca^2+^ signals in a more extensive array of cancer types.

## 4. TRPA1-Mediates ROS-Dependent Intracellular Ca^2+^ Signals in Cancer Cells: Survival vs. Apoptosis

Redox signaling has long been known to regulate cellular fate via distinct spatio-temporal Ca^2+^ signatures [17,64,101,102]. Low-to-moderate ROS levels can induce intracellular Ca^2+^ oscillations that regulate proliferation [103,104,105], gene expression [103,106], and mitochondrial bioenergetics [107,108], while excessive ROS production results in a continual and persistent rise in [Ca^2+^]_i_ that stimulates cell death [109,110]. Based on the evidence that TRPA1 presents a high redox-sensing capability and that intracellular Ca^2+^ signaling finely tunes tumorigenesis, recent investigations sought to unravel whether and how TRPA1 confers cancer cells the ability to cope with or succumb to oxidative stress.

### 4.1. TRPA1-Mediated Ca^2+^ Influx Promotes Cancer Cell Survival to Oxidative Stress

Takahashi and coworkers recently carried out a systematic investigation to assess the role of extracellular Ca^2+^ entry through TRPA1 in the engagement of an antioxidant defense program in breast and lung cancer cells [29]. This report showed that the TRPA1 transcript and protein were up-regulated in breast and lung tumors as compared to adjacent normal tissue. Furthermore, TRPA1 activation with another electrophilic agonist, i.e., mustard oil, induced a long-lasting increase in [Ca^2+^]_i_ that was abolished by the removal of extracellular Ca^2+^ and genetic (via a selective short hairpin RNA) or pharmacological (via AP-18) blockade of TRPA1 [29]. Intriguingly, the same approach demonstrated that TRPA1 mediated H_2_O_2_-evoked intracellular Ca^2+^ oscillations in breast and lung cancer cell lines [29]. In agreement with the pro-survival role of Ca^2+^ spiking in cancer cells [111,112], extracellular Ca^2+^ entry via TRPA1 was required to promote cell survival in TRPA1-enriched cancer cells challenged with H_2_O_2_ [29]. In addition, ectopic expression of TRPA1 rescued cell survival and prevented apoptosis in H_2_O_2_-treated TRPA1-low-expressing cancer cells [29]. By using a more physiological context, the authors unveiled an increase in ROS production in the inner region of breast and lung cancer spheroids, which led to TRPA1-mediated Ca^2+^ entry and resistance to oxidative stress. Importantly, TRPA1 activation did not reduce ROS levels, thereby indicating that TRPA1 does not contribute to scavenging ROS but rather to engaging an antioxidative defense program [29]. Furthermore, TRPA1-mediated Ca^2+^ influx promoted resistance to anoikis [29], i.e., the mode of apoptotic cell death that may occur when cells detach from the extracellular matrix (ECM) and migrate to a distant point to metastasize [113]. ROS generation is crucial to induce anoikis, but TRPA1 activation underlay detachment-induced Ca^2+^ signals and anoikis resistance in the inner region of tumor spheroids without reducing intracellular ROS levels [29]. Additionally, TRPA1-mediated Ca^2+^ entry promoted breast and lung cancer cell resistance to ROS-producing chemotherapeutics, such as carboplatin, doxorubicin, and paclitaxel [29]. In agreement with in vitro findings, this study demonstrated that genetic or pharmacological blockade of TRPA1 retarded breast and lung cancer growth and suppressed chemoresistance in immunocompromised mice and confirmed that tumor cells were also exposed to higher oxidative stress in vivo. Significant levels of 8-hydroxyguanosine (8-OHdG) and 4-HNE, which are common readouts of oxidative stress, were detected in cancer cells [29]. For instance, the 4-HNE concentration can increase up to the low micromolar range in the tumor microenvironment with potential pro-apoptotic effects against cancer cells [8]. Notably, 4-HNE has long been known to stimulate TRPA1-dependent intracellular Ca^2+^ signals [114,115,116]. The authors then exploited a reverse-phase protein array to unravel the Ca^2+^-dependent effectors that mediate the antioxidant defense triggered by TRPA1. They reported that TRPA1-mediated extracellular Ca^2+^ entry recruits the Ca^2+^/calmodulin-dependent proline-rich tyrosine kinase 2 (Pyk2) [117], which in turn engages several pro-survival signaling pathways, such as RAS-ERK, phosphatidylinositol 3-kinase (PI3K)/protein kinase B (AKT), and mammalian target of rapamycin (mTOR), and increases the expression of the anti-apoptotic protein MLC-1 (Figure 3) [29]. Finally, the authors demonstrated that TRPA1 expression in breast and lung cancer cells was regulated by NRF2, which can therefore prevent ROS-induced apoptosis by inducing the expression of both canonical (e.g., antioxidant) and non-canonical (e.g., TRPA1) oxidative stress defense proteins (Figure 3) [16,29]. Interestingly, mutations in *NFE2L2* and *KEAP1 genes*, which, respectively, encode for NRF2 and KEAP1, were associated with higher TRPA1 expression in lung tumors and head-neck squamous carcinoma [29]. Genetic silencing of NRF2 reduced TRPA1 expression in lung cancer cell lines, while it did not affect the expression levels of other TRP isoforms, such as TRPC3 and TRPV1 [29]. This observation is rather interesting since TRPV1 is also sensitive to ROS signaling [64,103,106], while TRPC3 is primarily regulated by diacylglycerol [118]. Therefore, NRF2 is likely to selectively control TRPA1 expression, although NRF2-dependent regulation of other ROS-sensitive TRP isoforms, such as TRPM2 [105] and TRPV4 [119], should also be investigated. Chromatin immunoprecipitation (ChIP)-coupled deep sequencing (ChIP-Seq) identified three putative NRF2-binding sites (Peak 1 to Peak 3) around the TRPA1 gene locus [29]. However, only NRF2 binding to thPeak1 region was able to induce TRPA1 expression [29].

A recent investigation suggested that H_2_O_2_-induced Ca^2+^ entry via TRPA1 could also engage an antioxidant defense program in melanoma, which presents a rather high oxidative stress in the tumor microenvironment [30]. The authors showed that the number of intratumoral and peritumoral M2 macrophages and the amount of 4-HNE progressively increased with tumor severity in cutaneous melanoma samples, while TRPA1 protein expression remained unchanged [30]. In addition, genetic (via a selective small interfering RNA) and pharmacological (via A967079) blockade of TRPA1 suppressed H_2_O_2_-evoked intracellular Ca^2+^ signals in the human melanoma cell lines WM266-4 and SK-MEL-28 [30]. Furthermore, TRPA1-mediated Ca^2+^ signals exacerbated H_2_O_2_-dependent ROS production [30], which could reflect the Ca^2+^-dependent recruitment of NOX4 [17]. Therefore, TRPA1 activation could amplify oxidative stress in cutaneous melanoma to amplify tumor progression [30]. Future work will have to assess whether TRPA1-induced ROS production delivers a pro-tumorigenic input or somehow tempers the efficacy of the non-canonical defense program engaged by TRPA1-mediated Ca^2+^ signals. Addressing this issue would be instrumental to better predicting the therapeutic outcome of TRPA1 manipulation (see also Section 5.3).

### 4.2. TRPA1-Mediated Ca^2+^ Influx Promotes ROS-Dependent Apoptosis in Cancer Cells

The findings described in [29] supported the prevailing view that extracellular Ca^2+^ entry via TRPA1 exerts an antioxidant defense effect in breast and lung cancer cells [16]. Nevertheless, parallel investigations provided evidence that the TRPA1-dependent increase in [Ca^2+^]_i_ may also support H_2_O_2_-induced apoptosis in other types of cancer cells. Temozolomide (TMZ) therapy represents the standard of care for the treatment of glioblastoma by inducing lethal DNA damage and subsequent ROS production [120,121]. Unfortunately, the development of TMZ resistance severely hampers its therapeutic efficacy and leads to patients’ death [122]. A recent report showed that TMZ induced the expression of O6-methylguanine DNA-methyltransferase (MGMT), a DNA repair enzyme that favors glioblastoma cell resistance, and MnSOD, an antioxidant gene, in the glioblastoma cell lines SHG-44 and U251 [28]. However, previous activation of TRPA1 with Compound 16a (PF-4840154) increased ROS production, enhanced apoptosis, and reduced MGMT/MnSOD expression, thereby reducing TMZ resistance [28]. Similar results were obtained by the ectopic expression of TRPA1 in U521 cells exposed to TMZ. Mechanistic analysis revealed that TRPA1-mediated Ca^2+^ entry boosted ROS production by exacerbating TMZ-dependent damage to mitochondrial dynamics [28]. Consistently, an independent study showed that hypoxia increased TRPA1-dependent membrane currents in another human glioblastoma cell line, i.e., DBTRG, thereby inducing cytosolic Ca^2+^ overload, mitochondrial depolarization, caspase-3 and caspase-9 activation, and apoptosis (Figure 4) [32]. TMZ is also employed to treat relapsed or refractory neuroblastoma [121]. Interestingly, TMZ induced apoptosis in SH-SY5Y neuroblastoma cells via the ROS-dependent activation of TRPA1 followed by mitochondrial dysfunction and caspase activation (Figure 4) [123]. Therefore, these findings indicate that TRPA1 stimulation could represent a promising therapeutic strategy to sensitize certain cancer types to ROS-induced apoptosis. This hypothesis has been supported by two recent investigations showing that AITC induced cytosolic Ca^2+^ overload and reduced viability in OSCC PE/CA-PJ41 cells [23], whereas cinnamaldehyde, another selective electrophilic TRPA1 agonist, induced ROS-dependent apoptosis in colon cancer cells [31].

### 4.3. Why Does ROS-Sensitive TRPA1-Mediated Ca^2+^ Influx Exert both Anti-Cancer and Pro-Tumorigenic Effects in Cancer Cells?

The highly heterogeneous spatio-temporal profile of Ca^2+^ signals orchestrates the recruitment of downstream Ca^2+^-dependent effectors [126] and determines whether an increase in [Ca^2+^]_i_ induces proliferation [111,127] or senescence [128,129], apoptosis [130,131,132] or autophagy [84,133], sensitivity [130,131] or resistance [125,134] to anticancer strategies. Therefore, the bimodal pro- and anti-oncogenic effect of ROS-dependent Ca^2+^ influx via TRPA1 in cancer cells is not surprising. Takahashi and coworkers demonstrated that TRPA1 triggers intracellular Ca^2+^ oscillations to promote ROS resistance in lung and breast cancers [29]. Conversely, preliminary evidence suggested that TRPA1 activation led to more protracted (and pervasive) Ca^2+^ elevations in OSCC [23] and glioblastoma [28,32], thereby resulting in cancer cell apoptosis. In accord, repetitive oscillations in [Ca^2+^]_i_ are nicely suited to recruit Ca^2+^-dependent effectors that promote cancer cell proliferation and survival, including Pyk2 [135,136,137,138], while avoiding mitochondrial Ca^2+^ overload [127,130,139,140,141]. It is unclear why TRPA1 activation evoked pro-oncogenic repetitive Ca^2+^ spikes in some, e.g., breast and lung, but not all tumor types that have been examined so far. Extracellular Ca^2+^ entry via other TRP channels, such as TRPM2 [142], TRPV1 [64], and TRPV4 [143], can elicit rhythmic Ca^2+^ release from the ER via Ca^2+^-induced Ca^2+^ release (CICR) through inositol-1,4,5-trisphosphate (InsP_3_) receptors (InsP_3_Rs). Future work should investigate whether TRPA1 channel protein is selectively coupled to ER-located InsP_3_Rs in lung and breast cancers rather than in OSCC, glioblastoma, and colorectal carcinoma. Furthermore, TRPA1 has also been detected in acidic lysosomal vesicles [37], which represent an emerging pro-oncogenic Ca^2+^ releasing organelle in cancer cells [84,127] and stimulate InsP_3_-dependent ER Ca^2+^ spikes via CICR [127,144,145]. Takahashi and coworkers showed that the removal of extracellular Ca^2+^ abolished H_2_O_2_-evoked, TRPA1-mediated intracellular Ca^2+^ oscillations in lung and breast cancer cells [29]. Nevertheless, the lysosomal expression of TRPA1 in cancer cells surely deserves future investigation.

## 5. Targeting TRPA1 to Sensitize Cancer Cells to Oxidative Stress

The bimodal effect of the TRPA1-mediated Ca^2+^ influx on the ability of cancer cells to cope or not with oxidative stress could be appropriately exploited for therapeutic purposes. The pharmacological blockade of TRPA1 could provide an effective strategy to reduce the Ca^2+^-dependent recruitment of antioxidant defense and pro-survival signaling pathways in lung and breast cancers, as recently shown in [146]. Conversely, stimulating TRPA1-mediated Ca^2+^ entry could represent a valuable tool to sensitize ROS-dependent cytosolic Ca^2+^ overload and apoptosis in OSCC, glioblastoma, and colorectal carcinoma. TRPA1 channels are the most broadly tuned chemosensory channels identified so far and are sensitive to a vast panel of small molecular drugs and natural compounds, which can either stimulate or inhibit their activity with a rather high selectivity [19,36,44].

### 5.1. TRPA1 Activators

TRPA1 channels can be stimulated not only by ROS but also by synthetic drugs, local anesthetics, environmental irritants, and plant-derived pungent compounds [19,36,44,147]. TRPA1 agonists can be broadly categorized as electrophilic compounds, which increase the open probability through covalent modifications, and non-electrophilic activators, which exert a non-covalent modulation of the channel. Electrophilic agonists stimulate TRPA1 activity by targeting the reactive thiol groups of cysteine and lysine at the NH_2_-terminal domain that are also able to sense intracellular ROS (see Section 2.5). These compounds include AITC, cinnamaldehyde, allicin, mustard oil, hydrogen sulphide, diallyl sulfide, acrolein, and JT010 [19,36,44,147,148]. Non-electrophilic modulators induce TRPA1-mediated Ca^2+^ entry without inducing covalent modifications of the channel protein and include menthol, thymol, carvacrol, local anesthetics (e.g., lidocaine, tetracaine, and procaine), anesthetic agents (e.g., propofol and etomidate), nonsteroidal anti-inflammatory drugs (e.g., acetaminophen), and several compounds exploited in cosmetics and therapeutics (e.g., alkyl esters of p-hydroxybenzoate or parabens) [19,36,44,147]. The mechanism whereby some non-electrophilic activators stimulate TRPA1 has been elucidated [44]. The non-covalent agonist, GNE551, was recently shown to increase the single-channel open probability of TRPA1 by interacting with a hydrophobic transmembrane binding site (Gln940) [149]. In contrast, the cell-penetrating peptidergic scorpion toxin (WaTx) can prolong the single-channel opening of TRPA1 by associating with an intracellular electrophile ligand-binding domain contributed to by Cys621 and Cys641 [150]. Therefore, a variety of agonists are potentially available to stimulate TRPA1 and enhance ROS-dependent apoptosis in cancer cell types that succumb in response to sustained Ca^2+^ entry via TRPA1. A comprehensive list of electrophilic and non-electrophilic TRPA1 agonists is presented in Table 1.

### 5.2. TRPA1 Blockers

The involvement of TRPA1 in a growing number of disorders (see Section 1) has favored the development of specific TRPA1 antagonists [19,36,44,147], which are listed in Table 2. The first and most widespread TRPA1 antagonist was synthesized by the Hydra Company based on its xanthine structure and was named HC-030031 [147]. Chembridge-5861528 is a derivative of HC-030031 that presents similar potency and specificity, but improved solubility [36,44]. However, the most potent available TRPA1 inhibitors are Compound 10, Compound 31, A-967079, and Glenmark 10, 15, 37 (synthesized by Glenmark), which display half-maximal inhibitory concentration (IC_50_) values within the nanomolar range [19,36,44]. In addition, TRPA1 activity is sensitive to the paracetamol analog (also termed acetaminophen) 6a/b, a novel battery of α-aryl pyrrolidine sulfonamides, and the analgesics tramadol and its metabolite M1 [19]. The *Sambucus ebulus* L. (SEB) fruit extract, which can exert beneficial therapeutic effects against inflammation and ER stress-related disorders [163], has recently been shown to inhibit TRPA1 [31]. However, SEB, as well as early TRPA1 inhibitors such as camphor and ruthenium red, can target other TRP channels and is therefore not selective [31]. On the other hand, the partial agonist AP-18 inhibits TRPA1-mediated Ca^2+^ entry by desensitizing the channel [36,44].

### 5.3. Could TRPA1 Manipulation Directly Affect Oxidative Stress in the Tumor Microenvironment?

A recent investigation demonstrated that H_2_O_2_-dependent TRPA1 activation can amplify oxidative stress in melanoma cell lines [30]. It is, however, still unclear whether TRPA1-induced ROS production delivers a pro-tumorigenic input or somehow tempers the efficacy of the non-canonical defense program engaged by TRPA1-mediated Ca^2+^ signals. Addressing this issue would be instrumental in better predicting the therapeutic outcome of TRPA1 manipulation. In accord, selective TRPA1 stimulation to promote caspase activation and cell death could exacerbate oxidative stress (e.g., in brain tumors), possibly further boosting cancer cell elimination. Nevertheless, if TRPA1-induced ROS production rather facilitates tumor progression, the therapeutic efficacy of this approach could be hampered by a negative-feedback mechanism. Conversely, blocking TRPA1-mediated Ca^2+^ signals to prevent the recruitment of non-canonical antioxidant programs (e.g., in lung and breast cancers) would limit TRPA1-induced ROS production, thereby either increasing (if TRPA1-dependent ROS are pro-tumorigenic) or tempering (if TRPA1-dependent ROS are pro-apoptotic) the therapeutic impact of TRPA1 manipulation.

## 6. Conclusions

Emerging evidence indicates that the Ca^2+^-permeable, non-selective cation channel TRPA1 is upregulated in cancer cells. TRPA1 is the most highly redox-sensitive TRP isoform and therefore its ability to mediate Ca^2+^ entry translates the high oxidative stress of the tumor microenvironment in an intracellular Ca^2+^ signal that can dramatically impact cancer cell fate. In breast and lung cancer, ROS-induced TRPA1 activation leads to intracellular Ca^2+^ oscillations that engage antioxidant and pro-survival signaling pathways, resulting in cancer cell tolerance to oxidative stress. TRPA1-mediated Ca^2+^ entry could therefore promote resistance to pro-oxidant therapies based on ROS-producing drugs, such as carboplatin, doxorubicin, and paclitaxel. In other solid malignancies, including glioblastoma and neuroblastoma, ROS-induced TRPA1 activation results in a persistent increase in [Ca^2+^]_i_ that causes mitochondrial Ca^2+^ overload and damage, thereby leading to caspase 3 activation and apoptotic cell death. These findings support the notion that the pharmacological manipulation of TRPA1-mediated Ca^2+^ signals could represent an alternative anticancer strategy. For instance, selective TRPA1 agonists (e.g., brain tumors) or blockers (e.g., lung and breast cancers) could be administered as adjuvant drugs of ROS-producing therapeutics, such as carboplatin, doxorubicin, and paclitaxel, to increase cancer cell sensitivity to oxidative stress.

## Figures and Tables

**Figure 1 cells-12-01261-f001:**
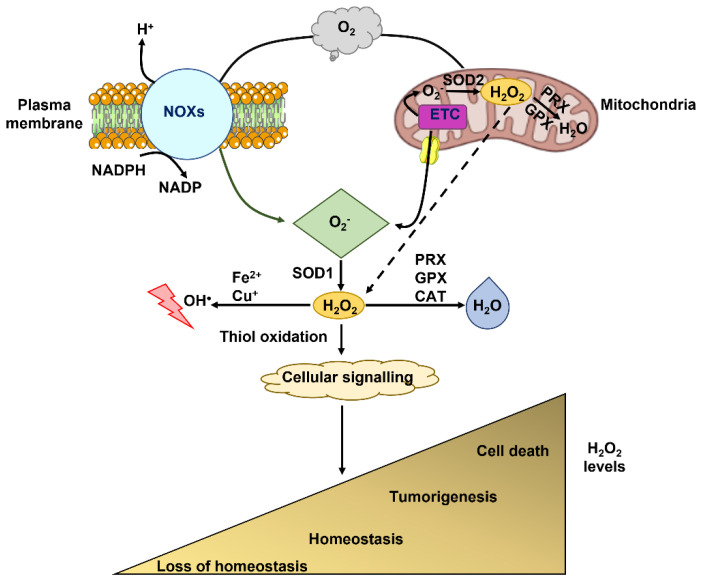
Generation and role of reactive oxygen species (ROS) in cancer cells. The membrane-bound NADPH oxidases (NOXs) and mitochondria are the main ROS generators in mammalian cells. NOXs catalyze the generation of intracellular superoxide anion (O_2_^•^) by operating the transfer of electrons from cytosolic NADPH to molecular oxygen (O_2_). Mitochondria produce ROS during cellular respiration: 1–2% of the electrons that are orderly transferred to the terminal electron acceptor, O_2_, via the electron transport chain (ETC) leak from the ETC and directly react with O_2_ thereby forming O_2_^•^. Mitochondria-derived O_2_^•^ can be released into the intermembrane space, thereby traversing the voltage-dependent anion channel into the cytosol. Herein, NOXs- and mitochondrial-derived O_2_^•^ are converted into hydrogen peroxide (H_2_O_2_) by cytosolic superoxide dismutase 1 (SOD1). Alternately, mitochondrial-derived O_2_^•^ is released into the mitochondrial matrix, where it is converted into H_2_O_2_ by the superoxide dismutase 2 (SOD2). H_2_O_2_ may freely diffuse across the mitochondrial membranes into the cytosol, or it can be detoxified into water (H_2_O) in the mitochondrial matrix by glutathione peroxidase (GPX) and peroxiredoxin (PRX). In the cytosol, H_2_O_2_ can stimulate cellular signaling via thiol oxidation of target proteins; it can be detoxified to H_2_O by PRX, GPX, and catalase (CAT); and it can interact with metal cations (Fe^2+^ and Cu^+^) to generate hydroxyl radical (OH^•^), which induce cellular damage by reacting with DNA, proteins, and lipids. A reduction in cytosolic H_2_O_2_ levels can disrupt cellular signaling and result in a loss of cellular homeostasis. Conversely, excessive H_2_O_2_ levels can lead to aberrant cellular signaling and favor tumorigenesis. Uncontrolled H_2_O_2_ levels can result in oxidative stress and cell death.

**Figure 2 cells-12-01261-f002:**
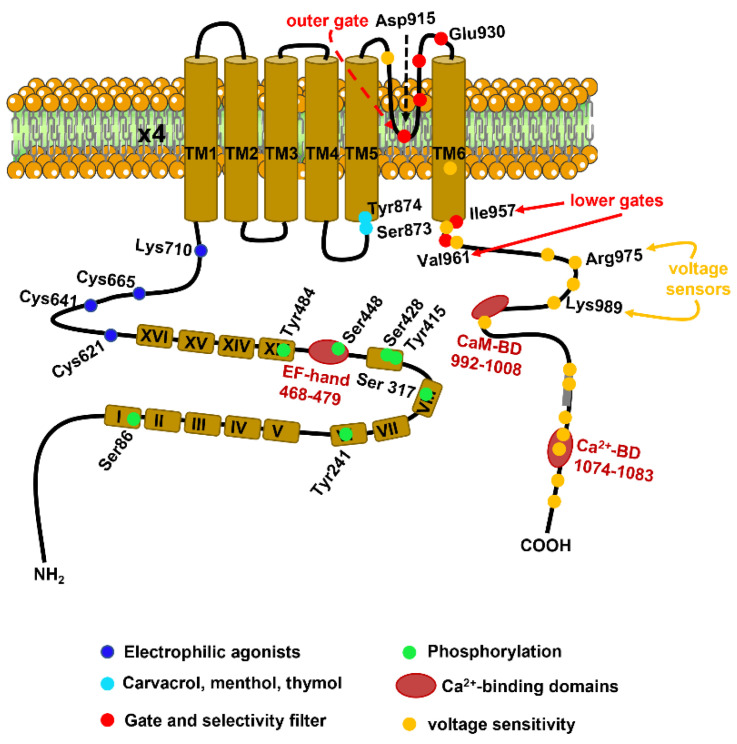
Molecular architecture of the human TRPA1 (hTRPA1) channel protein. Each hTRPA1 monomer comprises six transmembrane (TM) domains, a membrane-reentrant loop lining the channel pore between TM5 and TM6, and cytosolic NH_2_- and COOH-terminal tails. Brown rounded square shapes indicate the ankyrin repeat domains (ARDs), each indicated by a Roman numeral. ARD9 and ARD10 are not shown. Putative gates selectivity filter (red circles), voltage sensors (orange circles), and Ca^2+^-binding domains (BD) (dark red circles) are indicated. Single amino acid residues that regulate hTRPA1 channel function, underlie hTRPA1 modulation by various intracellular signaling pathways (e.g., phosphorylation and Ca^2+^-binding), or underlie agonist- or voltage-dependent gating are shown.

**Figure 3 cells-12-01261-f003:**
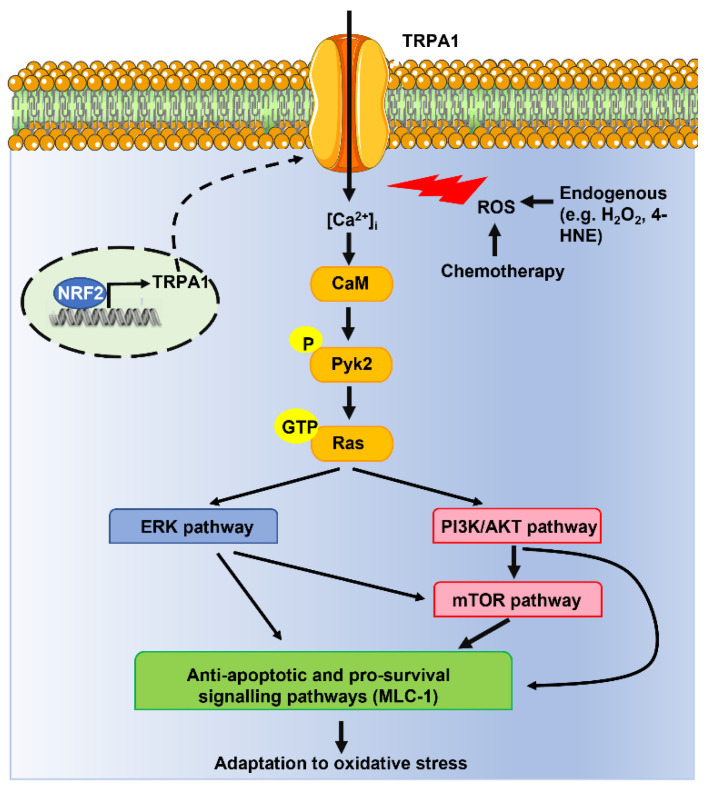
Extracellular Ca^2+^ entry via TRPA1 channels can induce adaptation to oxidative stress in cancer cells. ROS produced by tumor microenvironment (e.g., H_2_O_2_ and 4-HNE) or in response to chemotherapeutics (e.g., carboplatin, doxorubicin, and paclitaxel) activate TRPA1 on the plasma membrane. Extracellular Ca^2+^ entry, in turn, engages the Ca^2+^/CaM-dependent Pyk2, which stimulates the monomeric G-protein, RAS, to recruit several anti-apoptotic and pro-survival signaling pathways and thereby induce adaptation to oxidative stress. Furthermore, the redox-sensitive antioxidant transcription factor, NRF2, promotes TRPA1 expression, thereby triggering a positive feedback loop to enhance tolerance to oxidative stress and promote cancer cell survival.

**Figure 4 cells-12-01261-f004:**
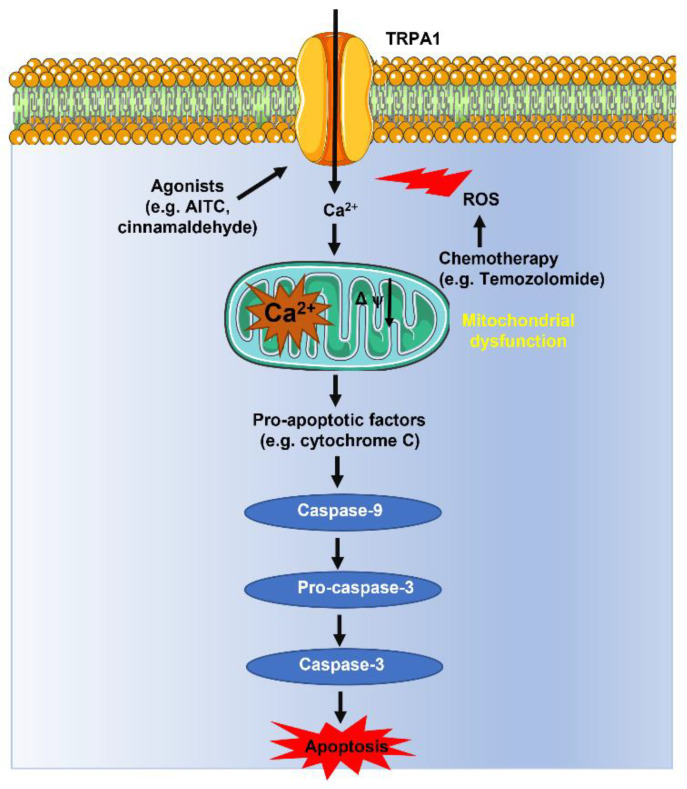
Extracellular Ca^2+^ entry via TRPA1 channels can induce mitochondrial Ca^2+^ overload, caspase-3 activation, and cell death in oxidative stress in cancer cells. ROS produced in response to chemotherapeutic treatments (e.g., Temozolomide) or selective agonists (e.g., AITC and cinnamaldehyde, but see also Table 1) can stimulate TRPA1 channels on the plasma membrane. The following influx of Ca^2+^ can induce mitochondrial Ca^2+^ overload and mitochondrial depolarization as well as the Ca^2+^-dependent assembly of the mitochondrial permeability transition pore (not shown), thereby releasing pro-apoptotic factors (e.g., cytochrome C and apoptosis-inducing factor, or AIF) into the cytoplasm [124]. Herein, cytochrome C interacts with apoptosis-activating factor 1 (Apaf-1, not shown) to form a supermolecular protein complex that recruits and activates the initiator caspase-9. Caspase-9, in turn, cleaves and activates the executioner, caspase-3 [125].

**Table 1 cells-12-01261-t001:** List of hTRPA1 activators.

Agonist	Source	Chemical Nature	EC_50_	Reference
Allyl isothiocyanate (AITC)	Mustard	Electrophilic	64 ± 3 µM	[73]
Cinnamaldehyde	Cinnamon	Electrophilic	400 ± 40 µM	[151]
Allicin	Garlic	Electrophilic	7.5 ± 0.4 µM	[152]
Hydrogen sulphide	Garlic	Electrophilic	1.8 ± 0.08 mM NaHS (mouse TRPA1)	[153]
Diallyl sulfide	Garlic	Electrophilic	254	[154]
Acrolein	Air pollutant	Electrophilic	5 ± 1 µM	[155]
JT-010	Synthetic	Electrophilic	0.047 µM	[156]
H_2_O_2_	ROS	Electrophilic	290 ± 90 µM	[157]
4-HNE	ROS	Electrophilic	5 µM	[158]
PGJ_2_	ROS	Electrophilic	5.6 µM (mouse TRPA1)	[65]
4-oxononenal (4-ONE)	ROS	Electrophilic	5.8 µM	[158]
4-hydroxyhexenal (4-HHE)	ROS	Electrophilic	≥4.3 µM	[158]
Menthol	Mint	Non-electrophilic	278 ± 30 µM	[159]
Thymol	Thyme	Non-electrophilic	127 µM	[160]
Carvacrol	Thyme	Non-electrophilic	7 µM	[160]
Lidocaine	Anesthetic	Non-electrophilic	24.000 ± 600 µM	[161]
Propofol	Anesthetic	Non-electrophilic	17 µM	[162]
WaTx	Synthetic	Non-electrophilic	16 µM	[150]

**Table 2 cells-12-01261-t002:** List of hTRPA1 inhibitors.

Inhibitor	IC_50_	Reference
HC-03031	5.3–6.2 µM	[164]
Chembridge-5861528	14.3–18.7 µM	[165]
AP-18	3.1 µM	[166]
A-967079	67 nM	[167]
Compound 10	170 nM	[168]
Compound 31	15 nM	[169]

## Data Availability

Not applicable.

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
