# Peer review of "Transient Receptor Potential Ankyrin 1 (TRPA1) Channel as a Sensor of Oxidative Stress in Cancer Cells"

_cells, 2023, doi:10.3390/cells12091261_

Round 1
Reviewer 1 Report
Excellent overview of the role of TPRA1 as a sensor of oxidative stress in cancer cells with clear pictures. Channel TPRA1 is upregulated in cancer cells. The pharmacological regulation of TPRA1-mediated Ca2+ signals as a therapeutic strategy to enhance cancer cells sensitivity to oxidative stress seems a promising new therapeutic pathway. I would like the authors to describe in more detail for which tumor types and at which point in the treatment they expect applications.
Reviewer 2 Report
This review provides a comprehensive overview of the structure and gating mechanisms of TRPA1, and explores the role of TRPA1-mediated intracellular calcium signaling in cancer cell proliferation, migration, and angiogenesis. The authors also discuss the redox-sensing capability of TRPA1 and how certain cancer cells may use it to engage a non-canonical antioxidant defense program, while ROS-dependent TRPA1 activation can lead to intracellular calcium overload, mitochondrial dysfunction, and apoptosis in other cancer types. Overall, this review sheds light on the relationship between TRPA1 and oxidative stress and presents a clear understanding of their role in tumor cells.
However, there are some areas that could benefit from further clarification or expansion. For example:
1. The review mentions that moderate levels of ROS can stimulate tumor development, while excessive levels can impair it. It would be helpful to define what is meant by "moderate" and "excessive" ROS levels or provide some examples of the ranges that have been observed in different types of cancer. Additionally, the authors could discuss how elevated ROS levels trigger the anti-oxidative mechanisms of tumors and provide some examples of these mechanisms.
2. The review briefly mentions that cancer cells have networks of antioxidant defense mechanisms to respond to oxidative stress. It would be useful to expand on this and discuss specific examples of these mechanisms.
3.The authors note that NRF2 is a master regulator of redox homeostasis in cancer cells but do not elaborate on its specific mechanism of action in regulating TRPA1 in tumor cells. This could be an area for further exploration.
4.The review provides a detailed description of the mechanism of action of ROS and TRPA1, but it would be helpful to add a discussion on the effect of drugs on the oxidative stress process mediated by TRPA1.
5. Finally, the authors could add their own insights and future outlook in each section, rather than simply presenting existing research results. This would help readers better understand the significance of the research presented and the potential implications for future studies.
Reviewer 3 Report
This is an important and well written contribution. Congratulations to the authors! The authors provided a very interesting review on TRPA1 as a sensor of oxidative stress. This topic is important and so far not sufficiently investigated, and may have high clinical significance for future anti-cancer strategies. The majority of the literature on TRPA1 is related to neurons and pain sensation. The assumption that TRPA1-mediated Ca2+ influx exerts both anti-cancer and pro-tumorigenic effects in cancer cells is challenging and needs future research.
Author Response
We are delighted by the Reviewer comments! We truly thank her/him for this very strong incentive to continue our work in this field. Thank you!